# VWF, Platelets and the Antiphospholipid Syndrome

**DOI:** 10.3390/ijms22084200

**Published:** 2021-04-18

**Authors:** Shengshi Huang, Marisa Ninivaggi, Walid Chayoua, Bas de Laat

**Affiliations:** 1Department of Functional Coagulation, Synapse Research Institute, 6217KM Maastricht, The Netherlands; s.huang@thrombin.com (S.H.); m.ninivaggi@thrombin.com (M.N.); w.chayoua@thrombin.com (W.C.); 2Cardiovascular Research Institute Maastricht, Maastricht University, 6229ER Maastricht, The Netherlands

**Keywords:** antiphospholipid syndrome, arterial thrombosis, von Willebrand factor, platelet, antiphospholipid antibody

## Abstract

The antiphospholipid syndrome (APS) is characterized by thrombosis and/or pregnancy morbidity with the persistent presence of antiphospholipid antibodies (aPLs). Laboratory criteria for the classification of APS include the detection of lupus anticoagulant (LAC), anti-cardiolipin (aCL) antibodies and anti-β2glycoprotein I (aβ2GPI) antibodies. Clinical criteria for the classification of thrombotic APS include venous and arterial thrombosis, along with microvascular thrombosis. Several aPLs, including LAC, aβ2GPI and anti-phosphatidylserine/prothrombin antibodies (aPS/PT) have been associated with arterial thrombosis. The Von Willebrand Factor (VWF) plays an important role in arterial thrombosis by mediating platelet adhesion and aggregation. Studies have shown that aPLs antibodies present in APS patients are able to increase the risk of arterial thrombosis by upregulating the plasma levels of active VWF and by promoting platelet activation. Inflammatory reactions induced by APS may also provide a suitable condition for arterial thrombosis, mostly ischemic stroke and myocardial infarction. The presence of other cardiovascular risk factors can enhance the effect of aPLs and increase the risk for thrombosis even more. These factors should therefore be taken into account when investigating APS-related arterial thrombosis. Nevertheless, the exact mechanism by which aPLs can cause thrombosis remains to be elucidated.

## 1. Introduction

The antiphospholipid syndrome (APS) is characterized by thrombosis and/or pregnancy morbidity with the persistent presence of antiphospholipid antibodies (aPLs). Laboratory criteria for the classification of APS include the detection of lupus anticoagulant (LAC), anti-cardiolipin (aCL) antibodies and anti-β2glycoprotein I (aβ2GPI) antibodies (Table 1) [1]. Both venous and arterial thrombosis can occur in APS patients, with venous thrombosis being slightly more prevalent. Ischemic stroke and transient ischemic attacks (TIAs) are the most common manifestations in the arterial circulation [2]. aPLs have been described to interfere with endothelial cells and the complement system, which may also be involved in the pathogenesis of arterial thrombosis [2].

The Von Willebrand Factor (VWF) VWF is a multimeric protein that circulates in a latent conformation, which can be converted into an active conformation. Active VWF is able to bind to the platelet-receptor glycoprotein-Iba (GPIba) and plays an important role in arterial thrombosis by mediating platelet adhesion and aggregation [4]. In healthy individuals, only a small portion of VWF circulates in its active conformation [5]. However, it has been shown that active VWF is increased in various pathological conditions, for example, Von Willebrand disease type 2B, congenital/acquired thrombotic thrombocytopenic purpura (TTP), HELLP-syndrome (hemolysis, elevated liver enzymes and low platelets), malaria and APS [6].

β2GPI is known to bind VWF and to inhibit platelet adhesion under flow conditions [4]. Moreover, aβ2GPI antibodies are able to inhibit the binding of VWF to β2GPI, thereby increasing the amount of active VWF present in the circulation. This indicates that β2GPI may act as a regulator of VWF-platelet interactions that are altered in the presence of aβ2GPI antibodies. This manuscript reviews existing literature on the role of VWF and platelets in the occurrence of arterial thrombosis in APS.

## 2. Von Willebrand Factor (VWF)

The VWF is a plasma glycoprotein of ca. 270 kDa (in the form of mature VWF) that is synthesized in megakaryocytes, platelets and endothelial cells (ECs). VWF plays a role in primary hemostasis and acts as an indicator for inflammation [7,8]. Another role is to bind factor (F) VIII, thereby protecting it from clearance, and to transport FVIII to the site of injury [9]. VWF is mostly known due to its role in several severe diseases such as von Willebrand disease (VWD) [10] and acquired von Willebrand syndrome (aVWS) [11]. Quantitative and/or qualitative abnormality in the adhesive plasma protein VWF will lead to VWD, one of the most common inherited bleeding disorders. In a recent study, Simurda et al. utilized plasma VWF carrying little FVIII and successfully inhibited VWD type III [12]. Thrombotic thrombocytopenic purpura (TTP) is a life-threatening disease due to a quantitative or qualitative defect in ADAMTS13 resulting in the increased presence of ULVWF, finally resulting in severe thrombotic events such as systemic platelet aggregation, organ ischemia [13].

As a multi-domain glycoprotein, VWF glycosylation occurs in both ER and Golgi apparatus [14]. Firstly, VWF co-translation folding occurs when entering the endoplasmic reticulum from the ribosome, while being accompanied by N-linked glycosylation mediated by oligosaccharyltransferase (OST) to promote correct protein folding. Secondly, N-linked and O-linked glycosylation presented in Golgi apparatus leads to the addition of sialylation, sulfation and blood group determinants to VWF multimers, affecting platelet adhesion, interaction with ADAMTS13 and VWF clearance [15,16,17,18].

The original product (pre-pro-VWF) coded by the *VWF* gene encompasses a signal peptide (SP), a VWF propeptide (VWFpp) and a mature VWF. A complete VWF monomer includes four types of homologous domains, which, in order from N- to C-terminal, are: SP -D1 -D2 -D’ -D3 -A1 -A2 -A3 -D4 -C1 -C2 -C3 -C4 -C5 -C6 -CK. In the endoplasmic reticulum (ER), the signal peptide is cut from the pre-pro-VWF and dimerization of pro-VWF monomers is followed by disulfide bonds near C-terminals [19,20]. VWF multimers are generated via N-terminal disulfide bonds in the Golgi apparatus through D1 and D2 domains supported by the VWF propeptide, with the process of VWF propeptide removal under the catalysis of furin. This process can lead to the development of ultra-large VWF multimers that can range from 500 kDa to more than 20,000 kDa [19,21]. After these biomodifications are done in the ER and Golgi apparatus, the mature VWF (VWF antigen (VWF: Ag)) and VWFpp are partly secreted into the plasma. However, most of them will be stored in the ultra-large (UL) form within the Weibel–Palade bodies of ECs or the α-granules of megakaryocytes and platelets [20,22,23]. Acute endothelial cell activation can be observed by measuring both VWFpp and VWF: Ag, knowing that the half-life of VWFpp is shorter [20].

Under normal conditions, VWF circulates in the plasma in a spherical conformation [24]. Upon vascular damage, VWF is released by ECs and binds via its A1 domain (to collagen type I, III, IV and VI) and A3 domain (to type I and III) which is present in the perivascular connective tissue of the damaged vessel wall [20,21,25,26,27]. The binding of VWF to collagen will induce a conformational change that results in a stretched conformation thereby exposing the A1 domain [28]. The VWF-A1 domain can interact with platelet receptor GPIbα [29], allowing adhesion of circulating platelets [30,31,32]. Moreover, UL-VWF is also released into the plasma from the ECs and platelets and will likewise participate in platelet adhesion and aggregation. The excessive presence of UL-VWF multimers may result in the development of thrombosis as, due to its larger size, UL-VWF has more binding sites for platelets and collagen when it is unfolded [33]. However, in normal conditions, the development of thrombosis is prevented by ADAMTS-13 (a disintegrin and metalloprotease), which is responsible for degrading UL-VWF into much smaller proteins via the cleavage site on the A2 domain of VWF [34].

In atherosclerotic lesions, VWF can act as a bridge between collagen and platelets, as well as between platelets themselves, thereby being a major contributor to the development of arterial thrombosis. A meta-analysis investigating the relationship between the levels of VWF and ADAMTS-13 with arterial thrombosis illustrated that high levels of VWF were associated with coronary heart disease and ischemic stroke, while ADAMTS-13 levels were lower in stroke patients than in controls and patients with coronary heart disease [35]. Pickens et al. used an *ADAMTS-13 gene* knock-out mouse model to verify the importance of ADAMTS-13 in arterial thrombus formation. The authors found that the thrombus formation velocity was significantly lower in transgenic mice that expressed recombinant human ADAMTS-13 compared to ADAMTS-13 knock-out mice but was similar to wild-type mice [36]. In accordance, Masias et al. showed a lower activity of ADAMTS-13 in patients with ischemic stroke and myocardial infarction, supporting the finding that ADAMTS-13 is involved in arterial thrombosis [37]. Moreover, Verhenne et al. hypothesized that the interaction between GPIbα and VWF is involved in the development of stroke [38]. Brait et al. found that a CD69 deficiency can promote thrombophilia by increasing the circulating plasma levels and activity of VWF [39]. In addition, in vivo experiments with DTRI-031, a novel aptamer designed by Nimjee et al. that inhibits VWF-mediated platelet adhesion, could inhibit platelet aggregation and thrombosis in mice [40].

Furthermore, the incidence of arterial thrombosis is significantly reduced in patients with VWD and there is increasingly more evidence that inflammation can cause VWF-mediated thrombosis [41]. Possible mechanisms could be the activation of the endothelium, secretion of VWF, assembly of hyper-adhesive VWF strings and fibers, reduced cleavage by ADAMTS13, as well as adhesion and deposition of VWF-platelet-thrombi in the vasculature [42].

## 3. Von Willebrand Factor and APS

Lindsey et al. described in 1993 a possible correlation between the presence of aPLs antibodies and VWF levels. IgG antibodies from patients with systemic lupus erythematosus (SLE) and APS were isolated and co-incubated with human umbilical vein endothelial cells. This resulted in a significantly higher secretion of VWF compared to incubation with IgG from controls. This finding suggests that IgGs from patients with SLE or APS are able to stimulate the secretion of vWF and therefore might play a role in the thrombotic events observed in these patients [43]. In addition, elevated VWF antigen levels were found in APS patients, together with impaired endothelial function and increased carotid intima-media thickness [44]. Since then, more and more studies have investigated the relationship between endothelial cell activation and aPLs antibodies.

### β2GPI and VWF

β2GPI is a highly glycosylated single-chain protein that is present in plasma and is able to bind anionic charged phospholipids such as CL or phosphatidylserine (PS) [45]. β2GPI is considered to be the main antigen in APS. Several studies showed that the levels of aβ2GPI IgG antibody correlate with the levels of active VWF. As mentioned previously, β2GPI is able to prevent the binding of VWF A1-domain to the GPIbα receptor on platelets, thereby also preventing platelet aggregation. Moreover, it was also shown that aβ2GPI antibodies found in APS patients can bind and form a complex with β2GPI, which will prevent the inhibitory function of β2GPI on platelets and potentially contribute to thrombosis [4]. This mechanism could contribute to the development of thrombosis and consumptive thrombocytopenia observed in APS patients. Another study demonstrated that aβ2GPI antibodies inhibit ADAMTS-13 activity, thereby suppressing the degradation of UL-VWF degradation in vivo, which in turn increases the risk of thrombosis [46]. 

## 4. Platelet Activation

Thrombus formation is a process that consists of two interdependent mechanisms that involve the activation of platelets and activation of the coagulation system. When a vessel wall is damaged, the collagen that is located in the extracellular matrix (ECM) will be exposed to the blood. As mentioned previously, VWF exposes its A1 domain under high shear or bound to collagen Figure 1A. Subsequently, this A1 domain binds to the GPIbα receptor on platelets, which will make the platelets roll over the collagen [29]. This rolling creates the ideal condition for platelet glycoprotein VI (GPVI) and integrin α2β1 to bind to the subendothelial collagen matrix tightly. Platelets adhering to the exposed collagen will become activated, change in shape, secrete their granule content and form an aggregate with other platelets. Additionally, platelets will recruit VWF [47], fibrinogen [48], fibrin and/or fibronectin [49] via integrin αIIbβ3, which in turn will also mediate platelet adhesion and activation [50,51]. Interestingly, the binding between platelet GPIbα and VWF is flexible and allows the platelet to translocate over VWF, while the binding between platelets via integrin αIIbβ3 with VWF and fibrinogen is more rigid [48].

The platelet plug formed can be consolidated via clot retraction which is mediated via the binding of αIIbβ3 on actin filaments [52]. 

Activated platelets release soluble agonists, for example, adenosine diphosphate (ADP), Thromboxane A2 (TXA2) and thrombin, which induce autocrine and paracrine platelet activation. A wide variety of receptors are expressed on the platelet surface, for example, G protein-coupled receptors (GPCRs), glycoprotein receptors, CLEC-2, Toll-like receptor 2 (TLR2) and platelet endothelial aggregation receptor 1 (PEAR1). Most of them are transmembrane receptors that are responsible for clot formation due to trauma or bleeding but also for signaling transmission and other physiological processes, for example, inflammatory reactions. These receptors also play an important role in pathological conditions as tumor metastasis and autoimmune diseases. The most abundant receptor present on the platelet surface is GPCR, which is a family consisting of seven-transmembrane (7TM) α-helices and a G protein [53,54,55]. To date, about 1000 unique GPCRs have been identified in the human genome, which participate in extracellular ligand recognition and intracellular signaling transduction [56]. GPCRs are expressed in hematopoietic and vascular tissues and mediate platelet activation, such as protease-activating receptors (PAR family), ADP receptors (P2Y1 and P2Y12), TXA2 receptors (TP-α), prostaglandin E2 receptors (EP family), adrenoreceptor (α2A) and serotonin receptor (5HT2A), as well as platelet inhibition, such as prostacyclin receptor (IP), adenosine receptor (AR) and prostaglandin D2 receptor (DP1 receptor) (Figure 1B) [57]. When the receptors that activate the platelets are stimulated, the content of the platelet granules is released, which in turn will lead to more platelets adhering, spreading and secreting their granule content. The propagation of the clot formation is ensured by the two pathways: the inside-out signaling pathway (i.e., when activated αIIbβ3 binds to its ligands, e.g., VWF and fibrinogen) and the outside-in signaling pathway (i.e., a series of intracellular signaling events that start after the platelet is activated) [58]. Sokol et al. explored 12 selected single nucleotide polymorphisms (SNPs) within seven specific genes in patients suffering from sticky platelets syndrome (SPS). They found that platelet endothelial aggregation receptor 1 (PEAR1) is associated with increased overall platelet aggregation and reduced responsiveness to aspirin and may act as a protective factor for DVT in patients with SPS type II which is stimulated by epinephrine [59].

## 5. Platelets Activation and APS

One of the functions of the platelets is to provide an anionic surface on which the clotting reactions can take place. aPLs antibodies that are present in APS patients are able to bind to platelet receptors, thereby enhancing the clotting response [60]. Biasiolo et al. showed that lysed platelets and gel-filtered platelets from anticoagulated APS patients with a quiescent APS did not contain aCL antibodies or LAC activity, which would mean that the antibodies do not bind to platelets in circulation, but that the binding only occurs after the platelet is activated [61,62].

In vivo and in vitro studies have shown that aPLs antibodies can affect platelet function. Forastiero et al. found an increased urinary excretion of 11-dehydro-thromboxane B2 in APS patients, which reflected in vivo platelet activation and correlated with aβ2GPI antibody levels [63]. Moreover, Espinola et al. reported that aPLs antibodies led to the enhancement of platelet activation by activating αIIbβ3 on the platelet surface [64]. As platelets play an essential role in the development of arterial thrombosis and cardiovascular disease in APS patients, patients at high risk for arterial thrombosis could benefit from treatment with platelet inhibitors such as clopidogrel (P2Y12 inhibitor) and seratrodast (TP antagonist).

### 5.1. Platelets and LAC

It has been shown that the expression of some platelet-derived proteins, such as β-thromboglobulin [65], platelet-derived extracellular vesicles [66] and soluble P-selectin [67], is upregulated in patients with LAC positivity. A platelet proteomics study conducted by Hell et al. revealed that LAC-positive patients (especially those who already had a thrombotic event) had altered protein profiles that were associated with a prothrombotic phenotype. This finding confirms that platelets contribute to the thrombotic risk of LAC-positive patients and could aid in finding an effective treatment for these patients [68]. Although the correlation between LAC and platelets has been investigated multiple times in the past, the underlying mechanism is still unclear.

### 5.2. Platelets and aβ2GPI

The binding of β2GPI-antibodies to β2GPI does not directly induce platelet activation, but there are receptors embedded on the phospholipid surface that are responsible for the interaction between β2GPI-aβ2GPI complex and the blood cells, such as TLR4 [69], ApoER2′ [70], GPIbα [70,71], TLR2 [72], TLR8 [73] and Fcγ-receptor IIa [74] (Figure 1B). In 1987, Nimpf et al. reported that β2GPI is not able to change the activation state of the platelets but rather prevents the secondary aggregation by inhibiting the release of dense granules, which is a prerequisite for ADP-induced secondary aggregation. This indicates again that the inhibiting effect of β2GPI is prevented when aβ2GPI antibodies are bound to β2GPI [75]. Fu et al. reported a significantly higher platelet aggregation and activation in patients with thrombocytopenia than those without thrombocytopenia, which might be associated with reduced β2GPI levels due to the presence of anti-β2GPI IgG antibodies [76]. Additionally, it was shown that the β2GPI-aβ2GPI complex induced platelet activation via GPIbα and apoER2’ which may then contribute to the prothrombotic tendency observed in APS patients. The ApoER2′ or GPIbα receptors could therefore be a potentially new and important therapeutic target for antithrombotic treatment in these patients [77]. Pathogenetic aβ2GPI have also been shown to correlate with abnormal activation of endothelium and monocytes by affecting the TLR4/NF-κB/MAPK pathway [78]. Moreover, He et al. reported that the β2GPI-aβ2GPI complex can promote an inflammatory response via the TLR4 signaling pathway, thereby accelerating the onset of atherosclerosis but delaying the accumulation of intima lipids [79]. Further research in mice demonstrated that the formation of the β2GPI-aβ2GPI complex increased the risk of abdominal aortic aneurysms through the binding on the TLR4 receptor [80]. Shoenfeld et al. hypothesized that the interaction between TLRs and aβ2GPI antibodies can promote the “second hit” that triggers the onset of thrombotic events in APS patients [81].

### 5.3. Platelets and aCL

The group of aCL antibodies consists of β2GPI-dependent and -independent antibodies. The β2GPI-dependent aCL antibodies require β2GPI as a co-factor for binding to CL and may, therefore, indirectly be dependent on the function and structure of β2GPI [82]. aCL is capable of promoting the synthesis of platelet-activating factors. This finding was first confirmed in 1991 by Silver et al., but it was not investigated whether other aPL could also be involved in this enhancement [83]. In 2002, Font et al. found that aβ2GPI-dependent IgM aCL antibodies can mediate the binding of platelets to collagen in the presence of β2GPI, thereby promoting the occurrence of thrombosis, which was not observed with β2GPI-independent IgM aCL antibodies [84]. Another study by Hollerbach et al. found that aβ2GPI-dependent IgG aCL antibodies could induce platelet binding to collagen, platelet aggregation and the release of p-selectin, which appeared to be mediated via the low-affinity Fcγ-receptor found on platelets [74].

The signaling pathways induced by those receptors on the surface of platelets mentioned above share the MAPK pathway, which leads to TXA2 production and granule secretion, including α-granules and dense granules. In general, phosphatidylinositol 4,5-bisphosphate (PIP_2_) can be hydrolyzed by phospholipase Cβ (PLCβ, activated by GPCR) and phospholipase Cγ (PLCγ, activated by GPVI and FcγIIa), leading to the generation of diacylglycerol (DAG) and 1,4,5-inositol trisphosphate (IP_3_). Both DAG and IP_3_ trigger diacylglycerol regulated guanine nucleotide exchange factor I (Cal-DAG-GEFI) generation and only DAG can induce several isoforms of protein kinase C (PKC) activation, both of which activate the MAPK pathway [85,86]. GPVI, FcγIIa, GPIbα and integrin (αIIbβ3, α2β1) can induce the activation of the PI3K/Akt pathway which triggers the MAPK pathway, Ca^2+^ mobilization upregulation and granule secretion [87]. Low-density lipoprotein (LDL) receptors, including TLR2, TLR4, TLR8 and ApoER2′, also activate the PI3K/Akt pathway and downstream MAPK pathway [88,89] (Figure 2).

## 6. Arterial Thrombosis and APS

Arterial thrombosis, which includes myocardial infarction and stroke, is a major clinical manifestation of APS. It results from the development of a thrombotic clot in the coronary and cerebral circulation, respectively, and often leads to the discovery of the autoimmune disease in young women [90]. Observational studies suggest that patients with aPLs, but without clinical symptoms of APS, may also have an increased risk for thrombosis [91]. It remains poorly understood which aPLs have a crucial effect on the development of arterial thrombosis.

### 6.1. LAC and Arterial Thrombosis

LAC is considered to be the main cause of arterial thrombotic events in APS patients. Saidi et al. showed that patients with a positive LAC had an elevated risk for stroke [OR (95% CI) = 8.1 (2.4–27.5)] [92]. LAC is also regarded as an important marker for APS with thrombotic events in young patients with primary Sjögren’s syndrome [93]. Additionally, a case-control study of urban and rural populations in Tanzania showed a strong association between LAC and stroke especially in young and middle-aged individuals [94]. Several case reports indicated that LAC positivity should be sought in arterial thrombosis, especially in young adults, and treated to prevent recurrence [95,96,97,98]. 

A cohort study of 141 patients showed that high levels of aCL correlated with the occurrence of arterial thrombosis but none of the patients with aCL and without LAC had arterial thrombosis [99]. A large multicenter population-based case-control study demonstrated that LAC is a major risk factor for arterial thrombotic events in young women, and this risk further increases in the presence of other cardiovascular risk factors [90]. Reynaud et al. designed a meta-analysis to explore which aPLs contributed the most to the occurrence of arterial thrombosis. The results showed that LAC, aCL, aβ2GPI, aPT and aPS were all significantly correlated with arterial thrombosis, but especially LAC and aCL had the highest odds ratio for arterial thrombosis compared to the other aPLs [100].

### 6.2. aβ2GPI and Arterial Thrombosis

It is commonly known that aβ2GPI IgG/IgM antibodies play an important role in the clinical manifestation of APS. However, as demonstrated by several research groups aβ2GPI IgA antibodies may also be associated with ischemic stroke [101,102,103,104,105]. Arad et al. found that aβ2GPI antibodies are not only a marker of APS but are also directly involved in the pathogenesis of thrombosis [106]. Both criteria and non-criteria aβ2GPI are associated with arterial thrombosis, while only criteria aβ2GPI, that is, IgA aβ2GPI is significantly associated with venous thrombosis [107]. Several studies have shown that IgA aβ2GPI is a risk factor for arterial and venous thrombosis. A significant correlation was found between positive isolated IgA aβ2GPI and arterial thrombosis (*p* < 0.001), venous thrombosis (*p* = 0.015) and thrombosis in general (*p* < 0.001). Moreover, on adjusting for risk factors such as age, smoking and obesity, the association between isolated IgA aβ2GPI and arterial thrombosis (*p* = 0.0003) and all thrombosis (*p* = 0.0003) remained significant [108]. Tortosa et al. suggested that compared to venous thrombosis, the existence of IgA aβ2GPI was better correlated with arterial thrombosis [109].

### 6.3. aCL and Arterial Thrombosis

Epidemiological studies of patients with acute non-hemorrhagic stroke have shown that the levels of aCL antibodies were significantly elevated even before onset. Urbanski et al. concluded that patients with isolated but persistent IgM antibodies were older in age at diagnosis but had a strong association with stroke [110]. The presence of IgM aPLs remained an independent risk factor for stroke, even after adjustment for other cardiovascular risk factors, such as smoking, diabetes and hypertension. A study conducted by Sammaritano et al. in 1997 demonstrated that it is mainly the aCL IgG2 isotype, that is, the major subclass of aCL, that was associated with arterial and/or venous thrombosis [84]. In addition, Robin et al. showed that β2GPI-dependent aCL IgG is an important predictor for future stroke in men [111]. Matyja et al. demonstrated that especially aCL IgG antibodies are associated with an increased risk of arterial thrombosis [112]. Recently, a Bayesian meta-regression analysis was used to assess the risk of recurrent coronary artery disease (CAD) according to the presence of IgG aCL and concluded that patients with CAD and elevated IgG aCL are at doubled risk of a recurrent major adverse cardiac event [113]. However, the underlying mechanism by which aCL antibodies can cause thrombosis remains to be elucidated.

### 6.4. aPS/aPT and Arterial Thrombosis

aPS/aPT antibodies may also contribute to arterial thrombotic events in APS patients. In 2004, Lopez et al. showed that the presence of aPS antibodies had a predictive value and was associated with arterial thrombosis, rather than venous thrombosis or pregnancy morbidity [114]. Additionally, some studies demonstrated a strong correlation between aPS/aPT antibodies and the prevalence of cerebral infarction [115,116]. A systematic review conducted by Sciascia et al. described a detailed analysis of the relationship between aPS/aPT and thrombotic events and concluded that the presence of aPS/aPT antibodies does increase the risk of arterial thrombosis [117].

## 7. Conclusions

Due to the physiological properties of VWF, platelet recruitment via VWF is more likely to appear in the region of high shear force, for example, in a coronary artery. When this recruitment of platelets leads to a blockage of the blood flow, it causes the formation of “platelet-rich thrombi”, which are also called “white thrombi”. On the contrary, when a clot is formed at a site of low shear rate, the formed clot is mainly due to fibrin formation and red blood cell recruitment, also known as “fibrin-rich thrombi” or “red thrombi”. The essential role that VWF and platelets play in arterial thrombosis has been studied extensively [118,119]. aβ2GPI antibodies, possibly together with aCL antibodies, can induce an overactivation of VWF in APS patients. It has been shown that aβ2GPI antibodies can counteract the inhibitory interaction of β2GPI on VWF. Because of this, VWF will tend to bind to GPIbα on platelets via its A1 domain, causing the platelets to aggregate. However, the mechanism behind the correlation between aPL antibodies and VWF remains unclear. Nearly all aPLs antibodies were reported to be associated with platelet activation.

To conclude, aPLs antibodies present in APS patients are able to increase the risk for arterial thrombosis by upregulating the plasma levels of active VWF and by promoting platelet activation. LAC had the highest odds ratio for arterial thrombosis compared to the other aPLs. In addition, inflammatory reactions induced by APS also provide a suitable condition for arterial thrombosis, mostly resulting in ischemic stroke and myocardial infarction. The presence of other cardiovascular risk factors, such as gender, age, smoking and obesity, can enhance the effect of aPLs and increase the thrombosis risk even more. These factors should therefore be taken into account when investigating arterial thrombosis caused by APS. Nevertheless, the exact mechanism by which LAC can cause thrombosis in APS remains to be elucidated.

## Figures and Tables

**Figure 1 ijms-22-04200-f001:**
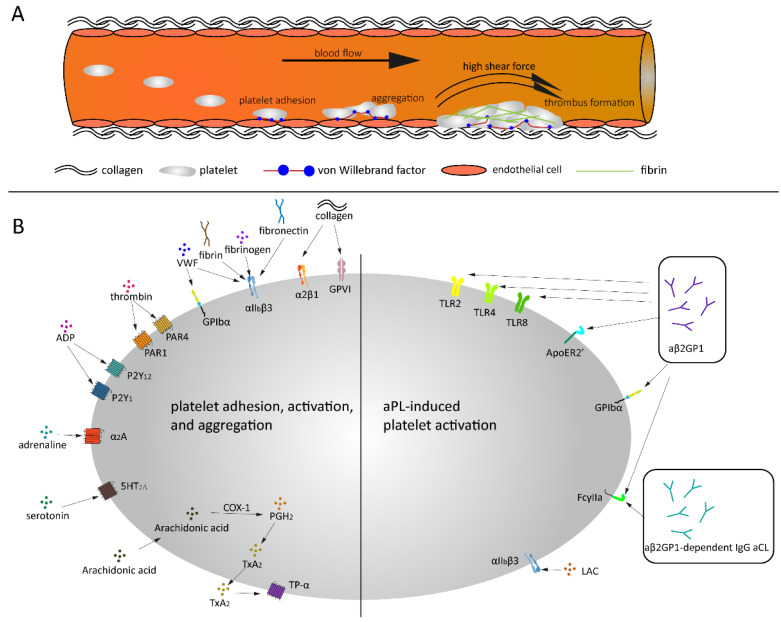
Thrombus formation under high shear and aPL related platelet activation. (**A**) Platelets bind to endothelial cells and collagen via VWF under high shear force, followed by platelets rolling over collagen and thrombus formation. (**B**) Under normal conditions, a variety of agonists and ligands bind their corresponding receptors, inducing platelet adhesion, activation and aggregation. aβ2GPI and aCL antibodies induce platelet activation in APS patients.

**Figure 2 ijms-22-04200-f002:**
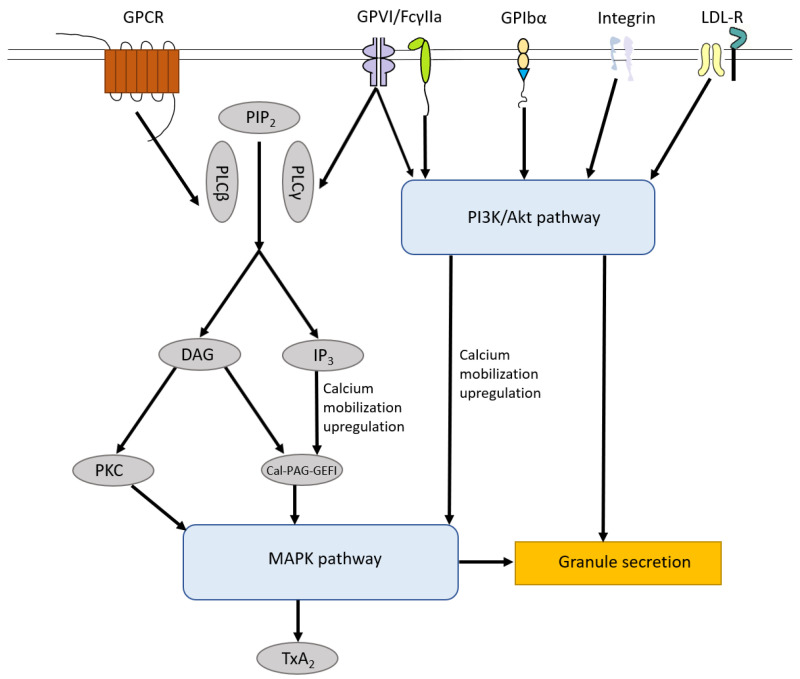
Platelet Signaling pathways. The signaling pathways induced by receptors on the surface of platelets including GPCR, GPVI, FcγIIa, GPIbα, integrin receptors and LDL-R share MAPK pathway, which leads to TXA2 production and granule secretion. In general, PIP_2_ can be hydrolyzed by PLCβ and PLCγ, leading to the generation of DAG and IP_3_. Both DAG and IP_3_ trigger Cal-DAG-GEFI generation and only DAG can induce numbers of isoforms of PKC activation, both of which activate the MAPK pathway. GPVI, FcγIIa, GPIbα and integrin (αIIbβ3, α2β1) can induce the activation of the PI3K/Akt pathway which triggers the MAPK pathway, Ca^2+^ mobilization upregulation and granule secretion. LDL-R, including TLR2, TLR4, TLR8 and Apo-ER2′, also activate the PI3K/Akt pathway and downstream MAPK pathway. PIP_2_, phosphatidylinositol 4,5-bisphosphate; PLCβ, phospholipase Cβ; PLCγ, phospholipase Cγ; DAG, diacylglycerol; IP_3_, 1,4,5-inositol trisphosphate; Cal-DAG-GEFI, diacylglycerol regulated guanine nucleotide exchange factor I; PKC, protein kinase C; LDL-R, Low-density lipoprotein receptors.

**Table 1 ijms-22-04200-t001:** Updated antiphospholipid syndrome (APS) classification criteria. APS is determined in the presence of at least one of the clinical criteria and one of the laboratory criteria. Adapted from Miyakis et al. [3].

**Clinical Criteria**
Vascular thrombosis: at least one clinical episode of arterial, venous or small-vessel thrombosis, in any tissue or organ; Pregnancy morbidity: ➢ at least one unexplained death of a morphologically normal fetus at or beyond the 10th week of gestation; ➢ at least one premature birth of a morphologically normal neonate before the 34th week of gestation because of eclampsia, severe preeclampsia or recognized features of placental insufficiency; ➢ no less than three unexplained consecutive spontaneous abortions before the 10th week of gestation, with maternal anatomic or hormonal abnormalities and paternal and maternal chromosomal causes excluded.
**Laboratory Criteria**
Lupus anticoagulant present in plasma, on two occasions at least 12 weeks apart; Anticardiolipin antibody of IgG and/or IgM isotype, in medium or high titer (>40 GPL or MPL or more than the 99th percentile), on two occasions at least 12 weeks apart; Anti-β2-glycoprotein-I antibody of IgG and/or IgM isotype, in medium or high titer (more than the 99th percentile), on two occasions at least 12 weeks apart.

## Data Availability

Not applicable.

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
