# Peer review of "VWF, Platelets and the Antiphospholipid Syndrome"

_ijms, 2021, doi:10.3390/ijms22084200_

Round 1

Reviewer 1 Report

Huang et al. provide a well written review on VWF, platelets and the antiphospholipid syndrome.

The only major point which needs to be addressed before publication is the description of VWF biosynthesis which is incorrect.

  • Line 63: What is meant by: “In the endoplasmic reticulum (ER), the pre-peptide is cut from the pre-pro-VWF”? The propeptide is cleaved of in the Golgi.
  • Line 64: VWF monomers are not generated in the Golgi. Dimers are formed in the ER and then dimers are multimerized in the Golgi. Dimerization occurs via disulfide bond formation between CK domains catalyzed by PDI in the ER, mulimerization occurs in the Golgi. Latter disulfides are formed between D’D3 domains supported by the propeptide.
  • It is mentioned that VWF is a glycoprotein in line 53 but glycosylation during biosynthesis is not further mentioned.
  • Line 71: “However, most of the secreted VWF will be stored in the ultra-large (UL) form within Weibel-Palade bodies”. The sentence is incorrect: the secreted VWF is secreted into the plasma and not stored in WPBs.
  • Line 75: As far as I know VWF binds collagen IV and VI via the A1 domain not the A3 (as also shown by Hoylaerts et al., 1997), please provide references for VWF binding sites for different types of collagen.

Minor points:

  • While the authors provide a good introduction on beta2GPI it is missing for cardiolipin. Please provide some background information on cardiolipin.
  • Line 46ff: A reference is missing for the statement “β2GPI is known to bind VWF and to inhibit platelet adhesion under flow conditions”.
  • Line 53: It should be stated that 270kDa is the molecular weight of mature VWF not the pre-pro-VWF.
  • Line 58: The statement on dysfunctional VWF in TTP needs to be specified. In TTP, VWF is not exactly dysfunctional but ultralarge VWF is enriched in plasma because is it not cleaved by ADAMTS13.
  • Line 79: “interacts” should read “interact”
  • Line 146: “platelets” should read “platelet”
  • In paragraph 6.2., the findings of the cited papers should be described a bit more. The paragraph is quite short and provides no information on the cited studies.

Author Response

We would like to thank all reviewers for their constructive comments. We feel that the manuscript has been greatly improved by their efforts.

Reviewer 1:

  1. Line 63: What is meant by: “In the endoplasmic reticulum (ER), the pre-peptide is cut from the pre-pro-VWF”? The propeptide is cleaved of in the Golgi.

Thanks for your suggestion. In this sentence, we’d like to use “pre-peptide” to refer to “signal peptide”, but it might make readers confused, so we changed it into “In the endoplasmic reticulum (ER), the signal peptide is cut from the pre-pro-VWF and dimerization of pro-VWF monomers is followed by disulfide bonds near C-terminals”. In addition, we added a new reference.

  1. Line 64: VWF monomers are not generated in the Golgi. Dimers are formed in the ER and then dimers are multimerized in the Golgi. Dimerization occurs via disulfide bond formation between CK domains catalyzed by PDI in the ER, mulimerization occurs in the Golgi. Latter disulfides are formed between D’-D3 domains supported by the propeptide.

Thanks for your suggestion on this site and it is indeed an incorrect description. We adopted your advice, but we thought VWF D1 and D2 domains support VWF multimerization. VWF D’-D3 stabilizes coagulation factor VIII (Kretz, C., & Yee, A. (2013). von Willebrand Factor: Form for Function. Seminars in Thrombosis and Hemostasis, 40(01), 017–027. doi:10.1055/s-0033-1363155) and support the formation of WBP Flood, V.H.; Schlauderaff, A.C.; Haberichter, S.L.; Slobodianuk, T.L.; Jacobi, P.M.; Bellissimo, D.B.; Christopherson, P.A.; Friedman, K.D.; Gill, J.C.; Hoffmann, R.G.; et al. Crucial role for the VWF A1 domain in binding to type IV collagen. Blood 2015, 125, 2297-2304, doi:10.1182/blood-2014-11-610824., maybe we can discuss that. We have changed it into “VWF multimers are generated via N-terminal disulfide bonds in the Golgi apparatus through D1 and D2 domains supported by VWF propeptide, with the process of VWF propeptide removal under the catalysis of furin.”

  1. It is mentioned that VWF is a glycoprotein in line 53 but glycosylation during biosynthesis is not further mentioned.

Thanks for your suggestion. We have added the process of VWF glycosylation process to our manuscript:

“As a multi-domain glycoprotein, VWF glycosylation occurs in both ER and Golgi apparatus. Firstly, VWF co-translation folding occurs when entering the endoplasmic reticulum from the ribosome, while accompanied with N-linked glycosylation mediated by oligosaccharyltransferase (OST) to promote correct protein folding. Secondly, N-linked and O-linked glycosylation presented in Golgi apparatus leads to the addition of sialylation, sulfation and blood group determinants on VWF multimers, affecting platelet adhesion, interaction with ADAMTS13, and VWF clearance”.

  1. Line 71: “However, most of the secreted VWF will be stored in the ultra-large (UL) form within Weibel-Palade bodies”. The sentence is incorrect: the secreted VWF is secreted into the plasma and not stored in WPBs.

Thanks for your suggestion on this wrong detail, we’d like to express previous content “mature VWF (VWF antigen (VWF: Ag)) and VWFpp”, so we change it to “most of them”.

  1. Line 75: As far as I know VWF binds collagen IV and VI via the A1 domain not the A3 (as also shown by Hoylaerts et al., 1997), please provide references for VWF binding sites for different types of collagen.

Thanks for your advice on this part. VWF A1 and A3 domains are both important on binding to collagen types, and as you said, different types of collagen bind to specific domains. Therefore we have changed to “VWF is released by ECs and binds via its A1 domain (to collagen type I, III, IV, and VI) and A3 domain (to type I and III) which is present in the perivascular connective tissue of the damaged vessel wall” and also with the references.

Minor points:

While the authors provide a good introduction on beta2GPI it is missing for cardiolipin. Please provide some background information on cardiolipin.

  1. Line 46ff: A reference is missing for the statement “β2GPI is known to bind VWF and to inhibit platelet adhesion under flow conditions”.

Thanks for your suggestion. We have added the reference.

  1. Line 53: It should be stated that 270kDa is the molecular weight of mature VWF not the pre-pro-VWF.

Thanks for your suggestion. We have changed it to “VWF is a plasma glycoprotein of ca. 270 kDa (in the form of mature VWF)” to highlight that its mature form is 270kDa.

  1. Line 58: The statement on dysfunctional VWF in TTP needs to be specified. In TTP, VWF is not exactly dysfunctional but ultralarge VWF is enriched in plasma because is it not cleaved by ADAMTS13.

Thanks for pointing out this mistake. Indeed, TTP does not meet the characteristics of loss of functionality, so we have changed it:

“Thrombotic thrombocytopenic purpura (TTP) is a life-threating disease due to a quantitative or qualitative defect in ADAMTS13 resulting in the increased presence of ULVWF, finally resulting in severe thrombotic events such as systemic platelet aggregation, organ ischemia [13]”.

  1. Line 79: “interacts” should read “interact”

Line 146: “platelets” should read “platelet”

We have corrected them.

  1. In paragraph 6.2., the findings of the cited papers should be described a bit more. The paragraph is quite short and provides no information on the cited studies.

Thanks for your suggestion. The last article is not appropriate in this paragraph and the cited references are not enough. Therefore we added more contents in this part:

Both criteria and non-criteria aβ2GPI are associated with arterial thrombosis, while only criteria aβ2GPI, i.e. IgA aβ2GPI is significantly associated with venous thrombosis [107]. Several studies have shown that IgA aβ2GPI is a risk factor for arterial and venous thrombosis. A significant correlation was found between positive isolated IgA aβ2GPI and arterial thrombosis (P < 0.001), venous thrombosis (P = 0.015), and thrombosis in general (P < 0.001). Moreover, considering adjusting for risk factors such as age, smoking, and obesity, the association between isolated IgA aβ2GPI and arterial thrombosis (P = 0.0003) and all thrombosis (P = 0.0003) remained significant [108]. Tortosa et al. suggested that compared to venous thrombosis, the existence of IgA aβ2GPI was better correlated with arterial thrombosis [109].”.

Reviewer 2 Report

In their review, Huang et al., described the role of platelets by focusing on vWF-mediated pathways in antiphospholipid syndrome. The manuscript is well written. I have only a remark regarding Figure 2, which does not contain enough information, especially figure legend. The authors should indicate intracellular pathways as well, which are activated upon ligand binding and extracellular stimuli. Figure 1 is missing, if not please correct 2 to 1.

Author Response

We would like to thank all reviewers for their constructive comments. We feel that the manuscript has been greatly improved by their efforts.

Reviewer 2:

In their review, Huang et al., described the role of platelets by focusing on vWF-mediated pathways in antiphospholipid syndrome. The manuscript is well written. I have only a remark regarding Figure 2, which does not contain enough information, especially figure legend. The authors should indicate intracellular pathways as well, which are activated upon ligand binding and extracellular stimuli. Figure 1 is missing, if not please correct 2 to 1.

Thanks for your suggestion and it is a very constructive advice. We added a paragraph to describe intercellular signalling pathways induced by different types of receptors:

“The signaling pathways induced by those receptors on surface of platelets mentioned above share the MAPK pathway, which leads to TXA2 production and granule secretion, including α-granules and dense granules. In general, phosphatidylinositol 4,5-bisphosphate (PIP2) can be hydrolyzed by phospholipase Cβ (PLCβ, activated by GPCR) and phospholipase Cγ (PLCγ, activated by GPVI and FcγIIa), leading to the generation of diacylglycerol (DAG) and 1,4,5-inositol trisphosphate (IP3). Both DAG and IP3 trigger diacylglycerol regulated guanine nucleotide exchange factor I (Cal-DAG-GEFI) generation and only DAG can induce several isoforms of protein kinase C (PKC) activation, both of which activate MAPK pathway [85,86]. GPVI, FcγIIa, GPIbα, and integrin (αIIbβ3, α2β1) can induce the activation of PI3K/Akt pathway which triggers the MAPK pathway, Ca2+ mobilization upregulation, and granule secretion [87]. Low-density lipoprotein (LDL) receptors, including TLR2, TLR4, TLR8, and ApoER2’, also activate PI3K/Akt pathway and downstream MAPK pathway [88,89] (Figure 2).”

Reviewer 3 Report

The authors of the manuscript focused on antiphospholipid syndrome and the role platelets and von Willebrand factor that play key role in arterial thrombosis in this syndrome. This is a current topic in medicine, as antiphospholipid syndrome are very common acquired thrombophilic state in the population.

Antiphospholipid syndrome is a disorder that manifests clinically as recurrent venous or arterial thrombosis and/or fetal loss. The manuscript is well structured and detailed review article, some facts need to be supplemented and corrected.

Page 2, lines 53-59: Patients with von Willebrand disease, not only impaired function of von Willebrand factor. The authors must state that VWD is the most common inherited bleeding disorder caused by a quantitative and/or qualitative abnormality in the adhesive plasma protein VWF. This statement was published in a manuscript which should be cited by Simurda et al. Successful Use of a Highly Purified Plasma von Willebrand Factor Concentrate Containing Little FVIII for the Long-Term Prophylaxis of Severe (Type 3) von Willebrand's Disease. Semin Thromb Hemost. 2017 Sep; 43 (6): 639-641.doi: 10.1055 / s-0037-1603362.

Page 4, lines 135-174:
In addition, the authors should state that platelets and platelet hyperagregability, intimately involved in the pathogenesis of thrombosis, can be activated by a variety of agonists through interactions with specific receptors localized on their membrane. Platelet hyperagregability, which is a disease in Sticky platelet syndrome (SPS) that is caused by genetic variants. Clinical symptoms of SPS include unexplained arterial and venous thrombotic events. Platelet endothelial aggregation receptor 1 is a type 1 membrane protein, which is expressed on platelets and endothelial cells. It has been shown that genetic variants in PEAR1 associate with increasing overall platelet aggregation and reduced responsiveness to aspirin in patients with premature cardiovascular disease. It is also appropriate to quote this publication: Sokol J et al. Association of Genetic Variability in Selected Genes in Patients With Deep Vein Thrombosis and Platelet Hyperaggregability. Clin Appl Thromb Hemost. 2018 Oct; 24 (7): 1027-1032. doi: 10.1177 / 1076029618779136. This study evaluated the variability of selected genes in a group of patients with SPS manifested as thrombosis. Several gene studies have examined the association of genetic variants in specific genes with platelet aggregation: (platelet endothelial aggregation receptor 1 [PEAR1], murine retrovirus integration site 1 [MRVI1], janus kinase 2 [JAK2], FCER1G, pro platelet basic protein [ PPBP], alpha2A adrenergic receptor [ADRA2A], and sonic hedgehog [SHH]).

Figure and table in the text are very clearly written.

Figure 2 is given in the text of the manuscript on page 6. Where is figure 1?

I have to say that with these 102 references there are only less than one-third of references newer than 5 years old. It is also appropriate to add newer references.

Author Response

We would like to thank all reviewers for their constructive comments. We feel that the manuscript has been greatly improved by their efforts.

Reviewer 3:

The authors of the manuscript focused on antiphospholipid syndrome and the role platelets and von Willebrand factor that play key role in arterial thrombosis in this syndrome. This is a current topic in medicine, as antiphospholipid syndrome are very common acquired thrombophilic state in the population.

Antiphospholipid syndrome is a disorder that manifests clinically as recurrent venous or arterial thrombosis and/or fetal loss. The manuscript is well structured and detailed review article, some facts need to be supplemented and corrected.

  1. Page 2, lines 53-59: Patients with von Willebrand disease, not only impaired function of von Willebrand factor. The authors must state that VWD is the most common inherited bleeding disorder caused by a quantitative and/or qualitative abnormality in the adhesive plasma protein VWF. This statement was published in a manuscript which should be cited by Simurda et al. Successful Use of a Highly Purified Plasma von Willebrand Factor Concentrate Containing Little FVIII for the Long-Term Prophylaxis of Severe (Type 3) von Willebrand's Disease. Semin Thromb Hemost. 2017 Sep; 43 (6): 639-641.doi: 10.1055 / s-0037-1603362.

Thanks for your constructive suggestion, we have added more details to our manuscript:

“Quantitative and/or qualitative abnormality in the adhesive plasma protein VWF will lead to VWD, one of the most common inherited bleeding disorders. In recently study, Simurda et al. utilized plasma VWF carrying little FVIII and successfully inhibited VWD type III”.

  1. Page 4, lines 135-174: In addition, the authors should state that platelets and platelet hyperagregability, intimately involved in the pathogenesis of thrombosis, can be activated by a variety of agonists through interactions with specific receptors localized on their membrane. Platelet hyperagregability, which is a disease in Sticky platelet syndrome (SPS) that is caused by genetic variants. Clinical symptoms of SPS include unexplained arterial and venous thrombotic events. Platelet endothelial aggregation receptor 1 is a type 1 membrane protein, which is expressed on platelets and endothelial cells. It has been shown that genetic variants in PEAR1 associate with increasing overall platelet aggregation and reduced responsiveness to aspirin in patients with premature cardiovascular disease. It is also appropriate to quote this publication: Sokol J et al. Association of Genetic Variability in Selected Genes in Patients With Deep Vein Thrombosis and Platelet Hyperaggregability. Clin Appl Thromb Hemost. 2018 Oct; 24 (7): 1027-1032. doi: 10.1177 / 1076029618779136. This study evaluated the variability of selected genes in a group of patients with SPS manifested as thrombosis. Several gene studies have examined the association of genetic variants in specific genes with platelet aggregation: (platelet endothelial aggregation receptor 1 [PEAR1], murine retrovirus integration site 1 [MRVI1], janus kinase 2 [JAK2], FCER1G, pro platelet basic protein [ PPBP], alpha2A adrenergic receptor [ADRA2A], and sonic hedgehog [SHH]).

Thanks for your suggestion. We have modified our manuscript:

“Sokol et al. explored 12 selected single nucleotide polymorphisms (SNPs) within 7 specific genes in patients suffered with sticky platelet syndrome (SPS). They found that platelet endothelial aggregation receptor 1 (PEAR1) is associated with increasing overall platelet aggregation and reduced responsiveness to aspirin and may act as a protective factor for DVT in patients with SPS type II which is stimulated by epinephrine”.

  1. Figure 2 is given in the text of the manuscript on page 6. Where is figure 1?

Thanks for your suggestion. The serial number of the picture is wrong. We have changed it.

  1. I have to say that with these 102 references there are only less than one-third of references newer than 5 years old. It is also appropriate to add newer references.

Thanks for your reminding. We have added more newer references to our manuscript in the process of editing the manuscript.

Round 2

Reviewer 3 Report

The presented manuscript has been corrected in response to the suggestions. The authors have followed the recommendations of the reviewer. After the revision, the provided data and addition of the results became more clear.  I would like to thank the authors for resubmitting the manuscript and explaining the obscure points from the previous version.